# High-throughput tracking enables systematic phenotyping and drug repurposing in *C. elegans* disease models

**Thomas J O'Brien[1,2†], Ida L Barlow[1,2†], Luigi Feriani[1,2], André EX Brown[1,2*]**

[1]Institute of Clinical Sciences, Imperial College London, London, United Kingdom;
[2]MRC London Institute of Medical Sciences, London, United Kingdom

## eLife Assessment

This **important** study provides proof of principle that *C. elegans* models can be used to accelerate the discovery of candidate treatments for human Mendelian diseases by detailed high-throughput phenotyping of strains harboring mutations in orthologs of human disease genes. The data are **compelling** and support an approach that enables the potential rapid repurposing of FDA-approved drugs to treat rare diseases for which there are currently no effective treatments. The work will be of interest to all geneticists.

**\*For correspondence:**
andre.brown@imperial.ac.uk

[†]These authors contributed equally to this work

**Competing interest:** The authors declare that no competing interests exist.

**Abstract** There are thousands of Mendelian diseases with more being discovered weekly and the majority have no approved treatments. To address this need, we require scalable approaches that are relatively inexpensive compared to traditional drug development. In the absence of a validated drug target, phenotypic screening in model organisms provides a route for identifying candidate treatments. Success requires a screenable phenotype. However, the right phenotype and assay may not be obvious for pleiotropic neuromuscular disorders. Here, we show that high-throughput imaging and quantitative phenotyping can be conducted systematically on a panel of *C. elegans* disease model strains. We used CRISPR genome-editing to create 25 worm models of human Mendelian diseases and phenotyped them using a single standardised assay. All but two strains were significantly different from wild-type controls in at least one feature. The observed phenotypes were diverse, but mutations of genes predicted to have related functions led to similar behavioural differences in worms. As a proof-of-concept, we performed a drug repurposing screen of an FDA-approved compound library, and identified two compounds that rescued the behavioural phenotype of a model of UNC80 deficiency. Our results show that a single assay to measure multiple phenotypes can be applied systematically to diverse Mendelian disease models. The relatively short time and low cost associated with creating and phenotyping multiple strains suggest that high-throughput worm tracking could provide a scalable approach to drug repurposing commensurate with the number of Mendelian diseases.

## Introduction

By definition, rare genetic diseases each affect a small number of patients. However, because there are many kinds of rare diseases, they are collectively common and affect more than 300 million people worldwide (*Nguengang Wakap et al., 2020*). Over 80% of rare genetic diseases affect children and many have severe symptoms (*Bavisetty et al., 2013*). Furthermore, 95% of rare diseases have no approved treatment (*Nguengang Wakap et al., 2020*). Given the severity and prevalence of rare diseases, the lack of treatments represents an important unmet need. Genome and exome sequencing

has become faster and cheaper, enabling its widespread use in the diagnosis of genetic diseases. In turn, this has led to the rapid identification of genetic lesions associated with rare genetic diseases (*Gonzaga-Jauregui et al., 2012*). Genetic diagnosis is an important first step, but often presents patients and physicians with a research problem rather than a treatment.

Genetic diagnoses rarely lead directly to a drug target hypothesis, especially for loss-of-function mutations. In the absence of a validated drug target, phenotypic screens in a model organism provide an alternative route to candidate treatments. Three things are required for a phenotypic screen to be successful: (1) a conserved gene; (2) a genetically tractable organism or culture system in which the disease mutation can be introduced; and (3) a measurable phenotype compatible with high-throughput screening. For the model nematode *C. elegans*, more than half of the human genes associated with rare genetic diseases in the OMIM database are conserved (*Kropp et al., 2021*) and CRISPR-based editing has made the generation of in vivo models containing disease-relevant mutations faster and easier. However, the third condition, a screenable phenotype, remains a bottleneck for many mutations and models.

If all three conditions are met, a phenotypic screen for candidate treatments can be performed. Because there is limited investment in drug development for rare diseases, lead optimisation and safety testing may not be possible. Hence, there is a particular focus on screening-approved drugs that have already been shown to be safe and bioavailable in humans to reduce the cost and time it takes to translate screening hits to human trials (*Roessler et al., 2021*). Two recent examples of promising repurposing screens using multiple model organisms illustrate this potential.

Firstly, modelling amyotrophic lateral sclerosis (ALS) with disease-associated genetic variants of conserved genes in *C. elegans*, zebrafish, and mice led to clinical trials for repurposing of the antipsychotic drug pimozide. By exploiting high throughput drug screening in *C. elegans* carrying point mutations in *TDP-43*, a panel of drugs was identified that rescued ALS movement and neurodevelopment phenotypes in all animal models. Early clinical trials found pimozide improved clinical and physiological outcomes in ALS patients (*Patten et al., 2017*). Second, the glycosylation disorder, PMM2-CDG, is caused by mutations in *PMM2* and presents clinically with developmental delay, psychomotor retardation, and axial hypotonia, among other symptoms. *C. elegans* carrying the same disease-associated mutation in their endogenous *pmm-2* gene exhibited larval arrest upon pharmacological ER stress. In vivo screening identified two compounds that rescued development in *C. elegans* and enzymatic activity in PMM2-CDG patient fibroblasts (*Iyer et al., 2019*). One of the compounds, the aldose reductase inhibitor epalrestat, is currently being tested in clinical trials for PMM2-CDG patients (National Clinical Trial number: NCT04925960).

A screenable phenotype was critical for the success of these studies. In the ALS model, worms became paralysed after 2 hr in liquid and in the PMM2-CDG model, brood size was severely affected. The majority (~74%) of rare genetic diseases affect nervous system function (*Lee et al., 2020*), and many of these mutations will not cause strong effects on development or gross motility when mutated in worms. High-content phenotyping that measures multiple features simultaneously could be useful in these cases to identify differences between strains that are not captured with univariate measures. In this study, we use a combination of high-throughput imaging (*Barlow et al., 2022*) and quantitative phenotyping (*Javer et al., 2018a*; *Javer et al., 2018b*) to phenotype a diverse panel of 25 *C. elegans* strains with mutations in orthologs of human disease genes. All but two of the mutant strains had detectable phenotypes compared to controls in at least one aspect of their morphology, posture, or motion. No single feature was different in all strains highlighting the need for a multidimensional phenotype. Mutation of genes predicted to have similar functions (e.g. *bbs-1*, *bbs-2*, and *tub-1*) led to similar phenotypes. As a proof-of-concept, we then performed a repurposing screen using a library of 743 FDA-approved compounds to identify drugs that improved the behavioural phenotype of *unc-80* loss-of-function mutants. Liranaftate and atorvastatin rescued the core behavioural features associated with *unc-80* loss of function and did not cause a large number of detectable side effects.

The ability to detect phenotypic difference in diverse strains using a standardised 16 min assay will make it possible to perform repurposing screens for existing and newly described rare diseases efficiently.

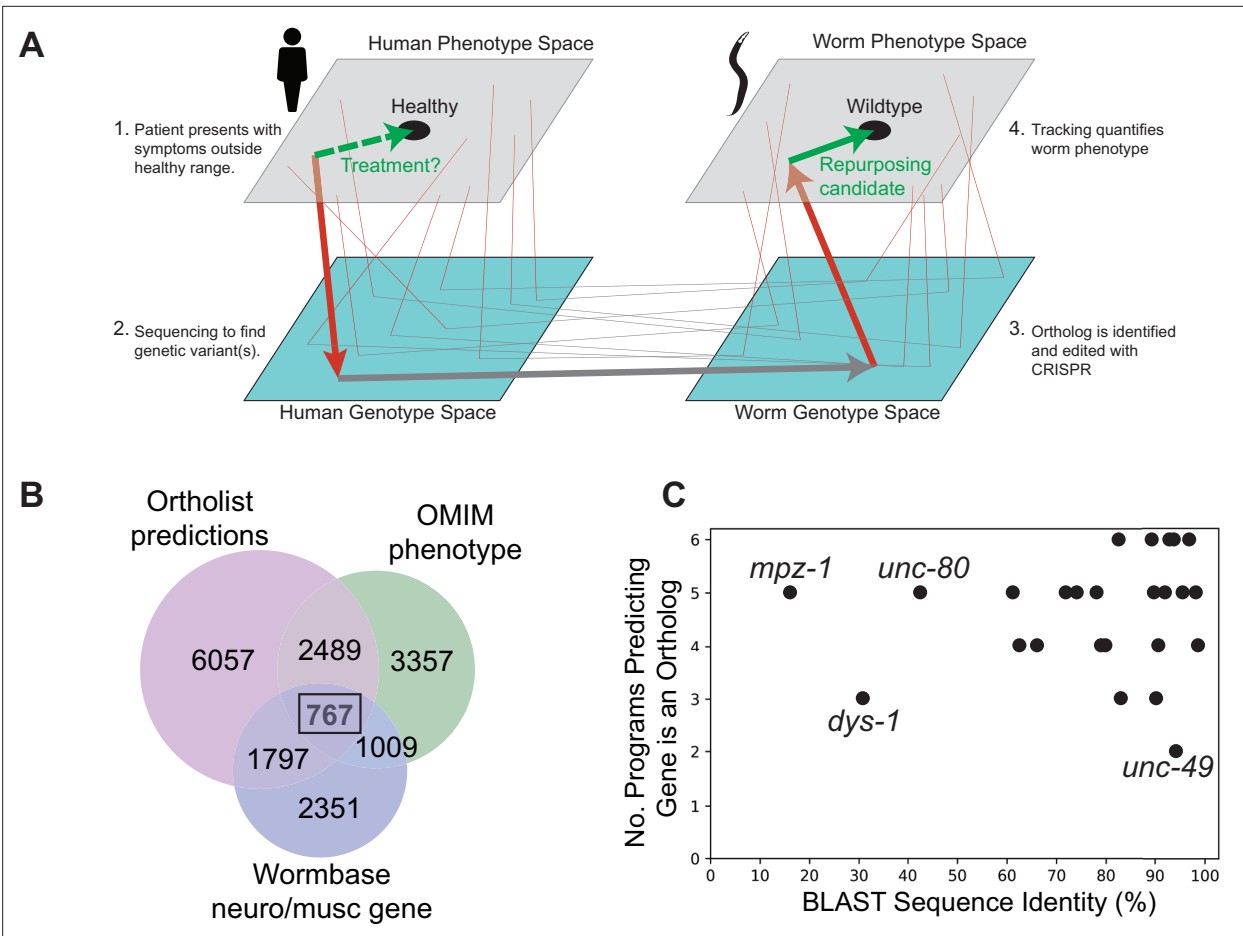

**Figure 1.** Overview of genotype-phenotype mapping and disease model panel. Disease modelling as genotype-phenotype mapping in humans and a model organism. Arrows show the progression from symptom identification (a phenotype outside the healthy range) to genotyping (placing a patient in genotype space by identifying a genetic variant), disease model creation (making a corresponding mutation in a model organism), and model organism phenotyping. Phenotypic drug screens identify compounds that move a disease model toward the wild-type phenotype (green arrow). These candidates can then be tested in humans (dashed green arrow). Thin lines show other symptom-gene-ortholog-phenotype connections. (**B**) Venn diagram showing a number of conserved genes (Ortholist 2), those involved in neuron or muscle function (Wormbase), and those associated with human genetic disorders according to the Online Mendelian Inheritance in Man (OMIM) database. (**C**) Sequence similarity between human and *C. elegans* genes and the total number of orthology programs predicting that the gene is an ortholog.

## Results

### Selection and molecular characterisation of a diverse panel of *C. elegans* disease models

The problem of large-scale disease modelling can be thought of as a problem of sampling and connecting two related genotype-phenotype (GP) maps (*Figure 1A*). The first is the human disease map. The genomic space is sampled by mutation and mating in humans and the map is elucidated by sequencing and clinical phenotyping of patients with genetic diseases. The second is the model organism map. Here, the genomic space is sampled using mutagenesis and genome editing and the map is elucidated using laboratory experiments. Given the accelerated characterisation of the human GP map, we sought to accelerate the characterisation of the corresponding *C. elegans* GP map. We follow a phenology-inspired approach to connecting the two maps. Provided there is a causal connection between a genetic variant and a disease in humans and that the causal gene is conserved in worms, the observed worm phenotype may be a useful disease model even if the connection to the human phenotype is non-obvious (*McGary et al., 2010*).

To identify a set of *C. elegans* genes that samples a broad genotypic space with connections to the human disease GP map, we first filtered the full database of human-*C. elegans* gene orthologs,

Ortholist 2 (*Kim et al., 2018*), using three criteria: (1) at least two orthology prediction algorithms agree the human and worm genes are orthologs; (2) the WormBase (version WS270) (*Harris et al., 2020*) gene description includes either 'neuro' or 'musc;' and (3) mutations in the human gene are associated with a Mendelian disease. A total of 767 *C. elegans* genes with 2558 orthologous human protein-coding genes met these criteria (*Figure 1B*). We used keyword searches to further reduce the list to 543 genes. Genes were kept if their WormBase gene description included any of the following: ion, ion channel, dopamine, serotonin, 5-HT, glutamate, acetylcholine, behaviour, behaviour, disease, epilepsy, autism, Parkinson, schizophrenia, bipolar, ADHD, seizure, GPCR, coupled receptor, antidepressant, antipsychotic. We prioritised genes associated with autosomal recessive disease in humans to increase the chance that a gene deletion would yield an appropriate model, and we focussed on those where loss-of-function mutants were known or likely to be viable. To get the list of 25 genes presented in this study, the final selection criterion was subjective interestingness based on our prior knowledge, WormBase gene descriptions, and/or brief literature searches.

In the final panel of 25 worm genes selected, 22 genes have >60% sequence similarity to their human ortholog, and 11 of the genes share >90% sequence similarity to their human counterpart (*Figure 1C*). Furthermore, 24/25 genes are predicted to be orthologous to human genes across >3 orthology prediction algorithms (*Figure 1C*), and all selected genes have a BLAST E-value (*McGinnis and Madden, 2004*) smaller than $3{\times}10^{-12}$ (*Supplementary file 1*).

Large CRISPR-Cas9 deletions (mean 4.4 kb) were made in each of the target genes to generate 25 strains containing a >55% deletion (average 76%) of the chosen gene. We direct the reader to the strain-specific gene cards (*Supplementary file 2*) for specific details on the size and position of genomic deletions for each of the individual mutants as well as a brief summary of the associated disease and worm phenotypes. Although we have focused on genes expected to have direct effects on neurons or muscles, even this small subset of disease-associated genes is predicted to affect diverse cellular processes including neurotransmission, excitability, development, and cellular structure (*Supplementary file 3*). According to OMIM, the mutated genes are associated with 31 rare genetic disorders in humans including intellectual disability, developmental delay, and disorders affecting the muscular and/or nervous systems, and are associated with >70 clinical presentations of disease according to the Human Disease Ontology database (*Schriml et al., 2022*; *Supplementary file 3*).

## Disease-associated mutations result in diverse phenotypes that are captured by multidimensional behavioural fingerprints

Given that there are thousands of rare diseases, methods to characterise the corresponding disease models should be as scalable as possible. Scalability is improved if a single standardised assay can be used to detect model phenotypes regardless of the mutation. High-resolution worm tracking and behavioural fingerprinting is a promising candidate for a widely applicable assay because it is both fast and multidimensional: a single minutes-long recording can capture differences in morphology, posture, and behaviour that can classify the effect of mutations and drug treatments (*Baek et al., 2002*; *Barlow et al., 2022*; *Geng et al., 2004*; *Javer et al., 2018b*; *McDermott-Rouse et al., 2021*; *Perni et al., 2018*; *Ramot et al., 2008*; *Restif et al., 2014*; *Swierczek et al., 2011*; *Tsibidis and Tavernarakis, 2007*; *Wang and Wang, 2013*; *Yemini et al., 2013*).

We tracked disease model worms on 96-well plates (*Figure 2A*) in three periods: (1) a 5 min baseline recording (prestim); (2) a 6 min video with three 10 s blue light stimuli separated by 90 s (blue light), and (3) a 5 min post-stimulus recording (poststim). For each recording, we extract 2763 features covering morphology, posture, and motion and concatenate the feature vectors together to represent the phenotype of each strain (8289 features in total). Many of these features are highly correlated, and so for clustering we use a pre-selected subset of 256 features (*Javer et al., 2018b*) (768 across the three recordings). Using these reduced feature vectors, we performed hierarchical clustering across all strains which demonstrates phenotypic diversity (many of the strains are both different from the N2 wild-type strain and from each other) as well as some similarities. For example, *bbs-1*, *bbs-2*, and *tub-1* mutants affect cilia formation and result in similar phenotypes that cluster together. The same is true for *kcc-2*, *snf-11* and *unc-25* mutants, which all affect GABAergic function. An interactive heatmap containing all behavioural feature vectors extracted for every disease model strain is also available for download (https://doi.org/10.5281/zenodo.12684130).

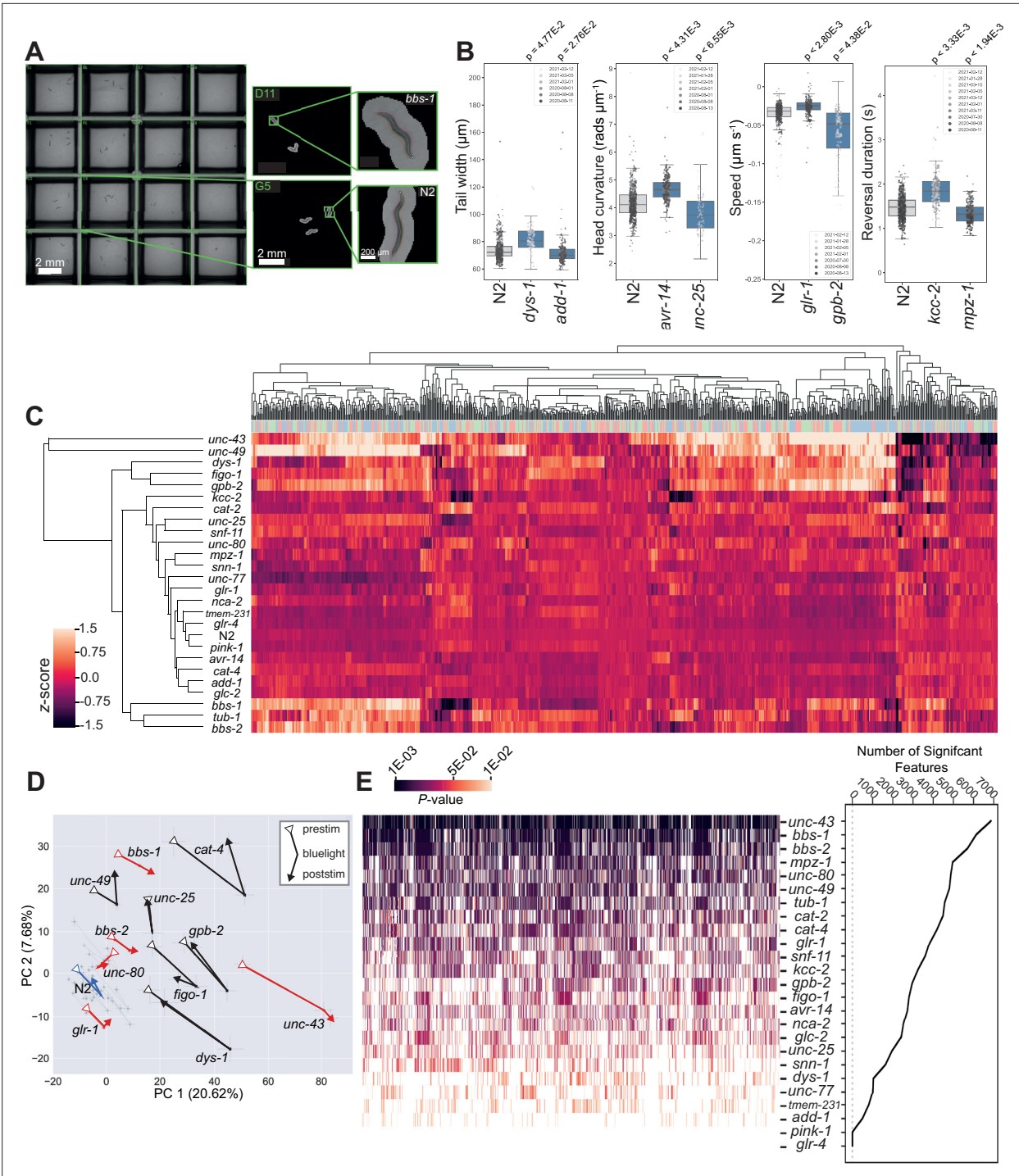

**Figure 2.** Diverse multidimensional behavioural phenotypes are obtained across the diverse panel of disease model mutants. (**A**) Representative field of view for a single camera channel with individual skeletonised worm images of *bbs-1(syb1588)* (top) and N2 (bottom) imaged on the same plate viewed using Tierpsy Tracker. (**B**) Representative behavioural phenotypes extracted by Tierpsy, representing changes in morphology, posture, and locomotion of different disease mutant strains. Boxes show interquartile range, error bars show minimum and maximum values excluding outliers. p-values are for comparisons to wild-type N2 worms, block permutation t-test, corrected for multiple comparisons. (**C**) Hierarchical clustering of behavioural fingerprints. Features are Z-normalised. The top barcode shows the period of image acquisition where the behavioural feature was extracted by Tierpsy: pre-stimulation (pink), blue light (blue), and post-stimulation (green). (**D**) Principal component analysis of the disease model mutants and N2 reference (blue). Strains move in phenospace between pre-stimulus (circular points), blue light (crosses) and post-stimulation (squares) recordings. Strains with an aberrant blue light response are shown in red. Error bars represent the standard error of the mean. (**E**) The total number of statistically significant behavioural features for each strain compared to N2 (from the total set of 8289 features extracted by Tierpsy). p-values for each feature were calculated

*Figure 2 continued on next page*

*Figure 2 continued*

using block permutation t-tests, using n=100,000 permutations, and p<0.05 considered statistically significant after correcting for multiple comparisons using the Benjamini-Yekutieli method.

Principal component analysis (PCA) of the same data, but now separated into prestimulus, blue light, and poststimulus recordings again show the separation of many of the mutant strains from the wild-type N2 (*Figure 2D*). The effect of the blue light stimulus on behaviour is clear from the motion of the points through this phenotype space and also reveals new phenotypes. While most strains show a partial recovery in the post-stimulus recording, some strains including *bbs-1*, *bbs-2*, *glr-1*, and *unc-43* display a sustained photophobic response (*Figure 2D*).

From the panel of 25 disease model strains, 23 are significantly different from the wild-type in at least one feature (*Figure 2E*). The strains that are most different from the wild-type show significant differences across a large number of features. These strains are particularly useful for high-throughput drug screens because the phenotypes can be reliably detected with a small number of replicates. Strain-specific summaries (gene cards) of each disease model mutant, its associated phenotype, and individual molecular characteristics are available in *Supplementary file 2*. Here, we provide a more detailed characterisation of two classes of mutants modelling ciliopathies and channelopathies in humans.

## Ciliopathies

Heritable mutations that affect cilia function are associated with a group of rare genetic disorders (ciliopathies) that share diverse symptoms including retinal dystrophy, developmental polydactyly, obesity, cognitive impairment, and renal dysfunction (*Reiter and Leroux, 2017*). The pleiotropic nature of ciliopathies is highlighted by significant interfamilial and intrafamilial phenotypic variability (*Shaheen et al., 2016*), and the resulting poor gene-phenotype correlation significantly complicates the discovery of effective therapeutics. As a result, current treatment regimens are based on treating individual symptoms.

In the longlist of 543 disease genes, there were 35 genes (6.4% of the total list) predicted to be involved in cilia function. We selected four genes (*bbs-1*, *bbs-2*, *tmem-231*, and *tub-1*) involved in regulating nonmotile (primary) cilia development and function that have confirmed expression in the cilia of amphid sensory neurons (*Taylor et al., 2021*; *Ward et al., 2008*; *Zhang et al., 2022*).

Of the genes affecting cilia function, two are orthologs of the Bardet-Biedl syndrome (BBS) family of genes (*bbs-1* and *bbs-2*). BBS is a pleiotropic syndrome often diagnosed in late childhood and is characterised by retinitis pigmentosa, cognitive impairment, obesity, renal dysfunction, hypogonadism, and polydactyly (*Forsythe and Beales, 2013*). The most prevalent BBS mutations are in *BBS1*, accounting for 23.4% of cases (*Forsyth and Gunay-Aygun, 2023*), and with 50% embryonic lethality and highly variable phenotypes, *Bbs1-/-* mouse models demonstrate model validity but have not yet yielded insights into potential therapies (*Forsyth and Gunay-Aygun, 2023*). Mutations in *BBS2* account for 8% of BBS cases (*Forsythe and Beales, 2013*) and are also implicated in other ciliopathies, such as Meckel syndrome (*Karmous-Benailly et al., 2005*).

Mutations in *bbs-1* and *bbs-2* led to strong phenotypes including changes in morphology, posture, and locomotion (*Figure 3A–D*). *bbs-1(syb1588)* and *bbs-2(syb1547)* mutants were shorter, wider, and had decreased curvature compared to wild-type worms (*Figure 3A, B and C*). They are also hyperactive, with a greater fraction of worms moving during pre-stimulus (baseline) recordings (see individual gene cards), and had faster body bends (a derivative of curvature) compared to N2 (*Figure 3D and F*). BBS proteins also regulate the *C. elegans* photoreceptor protein LITE1 in ASH sensory neurons through a DLK-MAPK signalling pathway independent of their function in cilia (*Zhang et al., 2022*). Consistent with this, we find that *bbs-1(syb1588)* and *bbs-2(syb1547)* mutants have attenuated blue light sensitivity, with both strains exhibiting a delayed forward, but enhanced backward photophobic escape response upon 10 s stimulation with blue light (*Figure 3F–H*).

The vertebrate family of tubby-like proteins (TUB, TULP) are required for correct GPCR localisation and intra-flagellar transport in primary cilia (*Mukhopadhyay and Jackson, 2011*). In humans, mutations in this family are associated with rod-cone dystrophy, obesity, and retinitis pigmentosa (*North et al., 1997*). Furthermore, *Tub1-/-* mice reflect the associated human disease pleiotropy, with insulin resistance, obesity, and cochlea and visual degeneration (*Sun et al., 2012*). *C. elegans* have a single

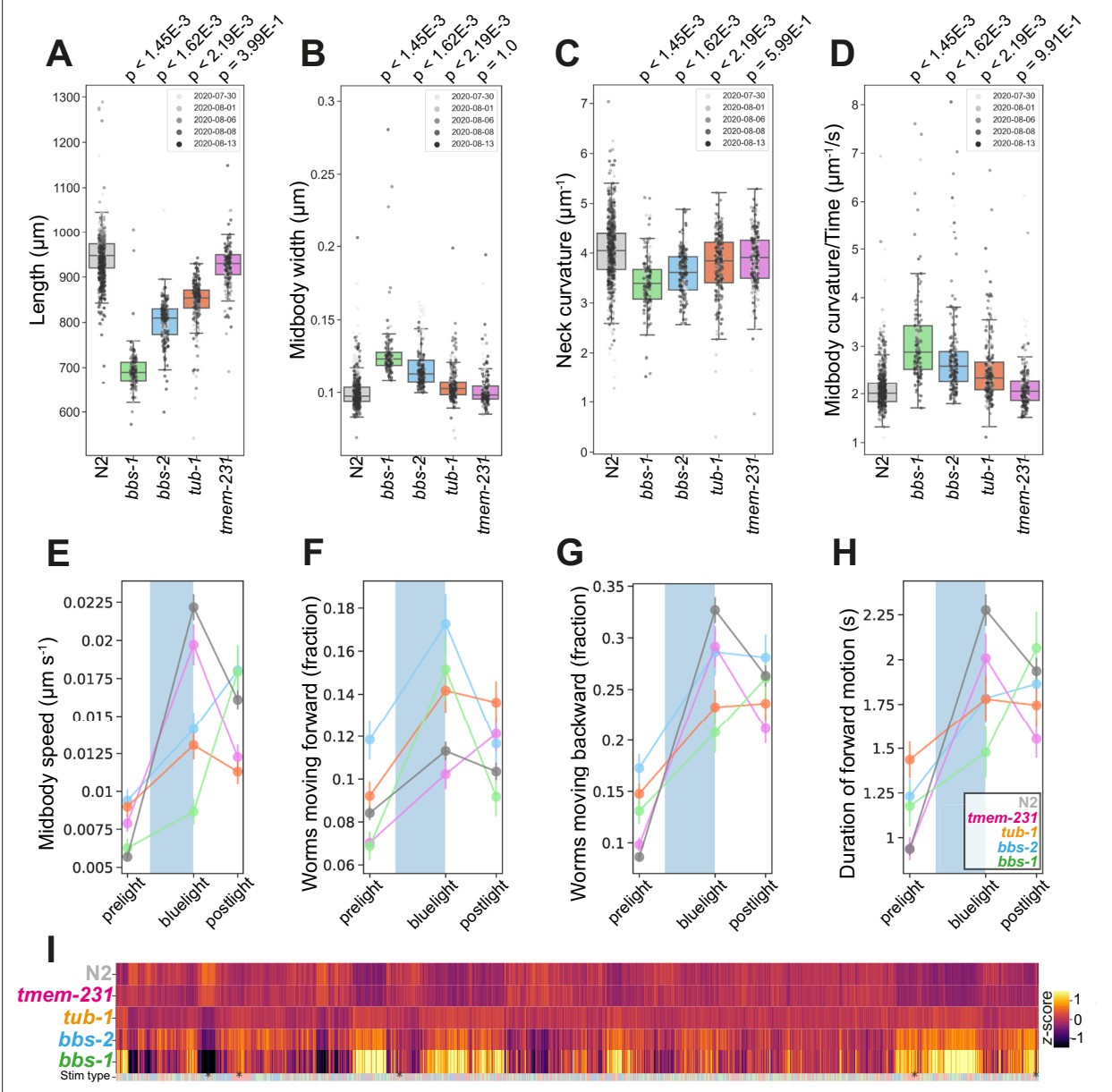

**Figure 3.** Disease model phenologs. (**A–D**) Key behavioural features altered in loss-of-function mutant strains associated with ciliopathies: *bbs-1(syb1588)*, *bbs-2(syb1547)*, *tub-1(syb1562)* and *tmem-231(syb1575)*, under baseline (pre-stimulus) imaging conditions. Individual points marked on the box plots are well-averaged values (three worms per well) for each feature across the independent days of tracking. Boxes show interquartile range, error bars show minimum and maximum values excluding outliers. p-values are for comparisons to wild-type N2 worms using block permutation t-tests (n=100,000 permutations, correcting for multiple comparisons using the Benjamini-Yekutieli method). (**E–H**) Changes in selected features in response to stimulation with a single 10 s blue light pulse (blue-shaded region). Feature values were calculated using 10 s windows centred around 5 s before, 10- s after, and 20 s after the beginning of each blue light pulse. (**I**) Heatmap of the entire set of 8289 behavioural features extracted by Tierpsy for the disease model strains associated with ciliopathies and N2. The 'stim type' barcode denotes when during image acquisition the feature was extracted: pre-stimulation (pink), blue light stimulation (blue), and post-stimulation (green). Asterisks show the location of selected features present in A-E.

ortholog of tubby-like proteins, *tub-1*, and mutants have defects in chemotaxis and insulin signalling as well as increased lipid accumulation (*Mak et al., 2006*).

*tub-1(syb1562)* mutants are shorter, wider, and hyperactive compared to wild-type worms, similar to *bbs-1* and *bbs-2* mutants (*Figure 3A, B and E*). They also have a defect in their response to blue light (*Figure 3F–H*). Unlike BBS mutants, cilia integrity is maintained in *tub-1* loss-of-function mutants (*Mak*

*et al., 2006*), which could explain the similar but less severe phenotype of *tub-1* mutants compared to *bbs-1* and *bbs-2* mutants.

Compartmentalisation of cilia-specific signalling components is regulated by a complex of transmembrane proteins including TMEM231 and mutations in this compartmentalisation complex are associated with neurodevelopmental limb defects and pathologies of the brain and kidneys found in Joubert and Meckel syndrome (*Chih et al., 2012*). Mouse *Tmem231-/-* mutations are embryonic lethal and the *C. elegans* ortholog *tmem-231* has been shown to have conserved molecular and cellular function (*Roberson et al., 2015*). However, unlike the other cilia-related mutants, *tmem-231* mutants do not have significant differences in body morphology or posture compared to N2 (*Figure 3*). We do note that *tmem-231(syb1575)* is slightly more active during baseline tracking (see strain-specific gene card).

## Channelopathies

The Na + cation leak channel (NALCN) is expressed throughout the central nervous system, and in parts of the endocrine (pancreas, adrenal, thyroid gland), respiratory and cardiac systems (*Cochet-Bissuel et al., 2014*). NALCN is a voltage-independent, nonselective cation channel that regulates resting potential (*Senatore et al., 2013*), and plays a role in neuromodulation by neurotransmitters (*Cochet-Bissuel et al., 2014*). Mutations in NALCN are associated with neuromuscular disorders including severe hypotonia, infantile neuroaxonal dystrophy (INAD), congenital contractures, cognitive delay, autism, epilepsy, bipolar disorder, and cardiac/respiratory problems (*Aoyagi et al., 2015*; *Cochet-Bissuel et al., 2014*; *Gal et al., 2016*). NALCN-knock-out mice die 12 hr after birth (*Lu et al., 2007*), so the development of appropriate non-mammalian animal models to understand the associated human diseases is essential.

*C. elegans* encodes two functionally redundant, but differentially expressed, NALCN homologs, NCA-2 and UNC-77, whose proper expression and axonal localisation in cholinergic neurons are regulated by UNC-80 (*Zhou et al., 2020*). Unlike murine models, *nca-2*, *unc-77*, and *unc-80* loss-of-function mutants are viable. Similar to *Drosophila melanogaster* or mouse neonates lacking the cation leak channel (*Cochet-Bissuel et al., 2014*), loss-of-function mutations in *nca-2* and *unc-80* result in morphological changes in *C. elegans*: both mutants are significantly shorter than wild-type N2 (*Figure 4A*).

Consistent with their differential expression (*Jospin et al., 2007*; *Yeh et al., 2008*), we observed phenotypic differences between *nca-2(syb1612)* and *unc-77(syb1688)* deletion mutants (*Figure 4*). Gain-of-function mutations in *unc-77* have previously been reported to cause deeper body bends (*Topalidou et al., 2017*) whereas *unc-77(syb1688)* deletion mutants have decreased curvature (*Figure 4B*). In contrast, *nca-2(syb1612)* has no significant change in curvature, but is slower than the wild-type strain (*Figure 4B–C*). We find that *unc-80(syb1531)* mutants, which affect the localisation of NALCN channel subunits, have the most severe phenotype exhibiting a decrease in both curvature and speed (*Figure 4B-C*).

Mutations in both NALCN channel subunits in *C. elegans* cause a 'fainter' phenotype when stimulated mechanically or immersed in liquid (*Pierce-Shimomura et al., 2008*). Similarly, we found that a ten-second blue light pulse resulted in increased post-exposure pausing and a decreased forward escape response in *nca-2(syb1612)* and *unc-80(syb1531)* single mutants (*Figure 4E–H*). *unc-77(syb1688)* mutants did not 'faint' after blue light exposure, but consistent with findings that the NALCN channel some regulates reversal behaviour (*Zhou et al., 2020*) both *unc-77(syb1688)* and *unc-80(syb1531)* fail to initiate a backward escape response. Pausing and 'fainting' after blue light exposure represent a novel screenable phenotype for NALCN-channelopathies.

## Drug repurposing screen in unc-80 mutants

Because *unc-80* mutants have a clear behavioural phenotype without a strong developmental phenotype, we reasoned that they were a useful test case for a drug repurposing screen using an acute 4 hr treatment which might rescue behavioural 'symptoms' in fully developed animals. When screening compound libraries, it is not practical to perform a large number of replicates which would be necessary to detect subtle phenotypes and overcome the reduced statistical power that comes from correcting for multiple comparisons in a large feature set (*Figure 4—figure supplement 1*).

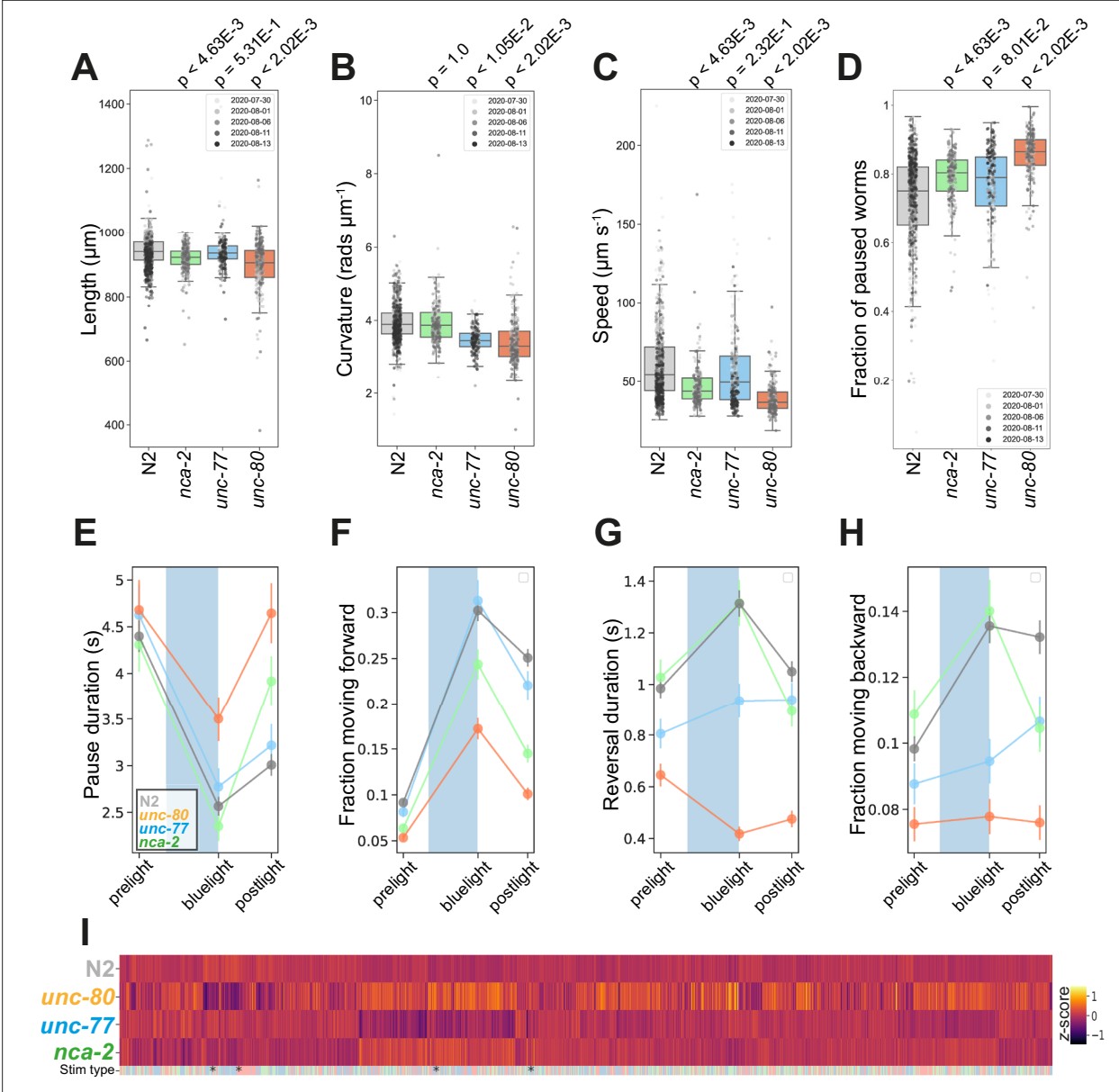

**Figure 4.** Na+ cation leak channel (NALCN) disease model phenologs. (**A–D**) Key behavioural and postural features altered in loss-of-function mutant strains associated with NALCN mutants: *nca-2(syb1612)*, *unc-77(syb1688)*, and *unc-80(syb1531)*, under baseline (pre-stimulus) imaging conditions. Individual points marked on the box plots are well-averaged values (three worms per well) for each feature across the independent days of tracking. Boxes show interquartile range, error bars show minimum and maximum values excluding outliers. p-values are for comparisons to wild-type N2 worms using block permutation t-tests (n=100,000 permutations correcting for multiple comparisons using the Benjamini-Yekutieli method). (**E–H**) Changes in selected features in response to stimulation with a single 10 s blue light pulse (blue-shaded region). Feature values were calculated using 10 s windows centred around 5 s before, 10- s after, and 20 s after the beginning of each blue light pulse. (**E**) A representative 'fainting phenotype' for *unc-80(syb1531)* and *nca-2(syb1612)*, characterised by an increase in pausing following the cessation of stimulation with blue light. (**I**) Heatmap of the entire set of 8289 behavioural features extracted by Tierpsy for the disease model strains associated with NALCN disease and N2. The 'stim type' barcode denotes when during image acquisition the feature was extracted: pre-stimulation (pink), blue light stimulation (blue), and post-stimulation (green). Asterisks show the location of selected features present in A-D.

The online version of this article includes the following figure supplement(s) for figure 4:

**Figure supplement 1.** Number of initial compound hits detected when analysing increasing numbers of features.

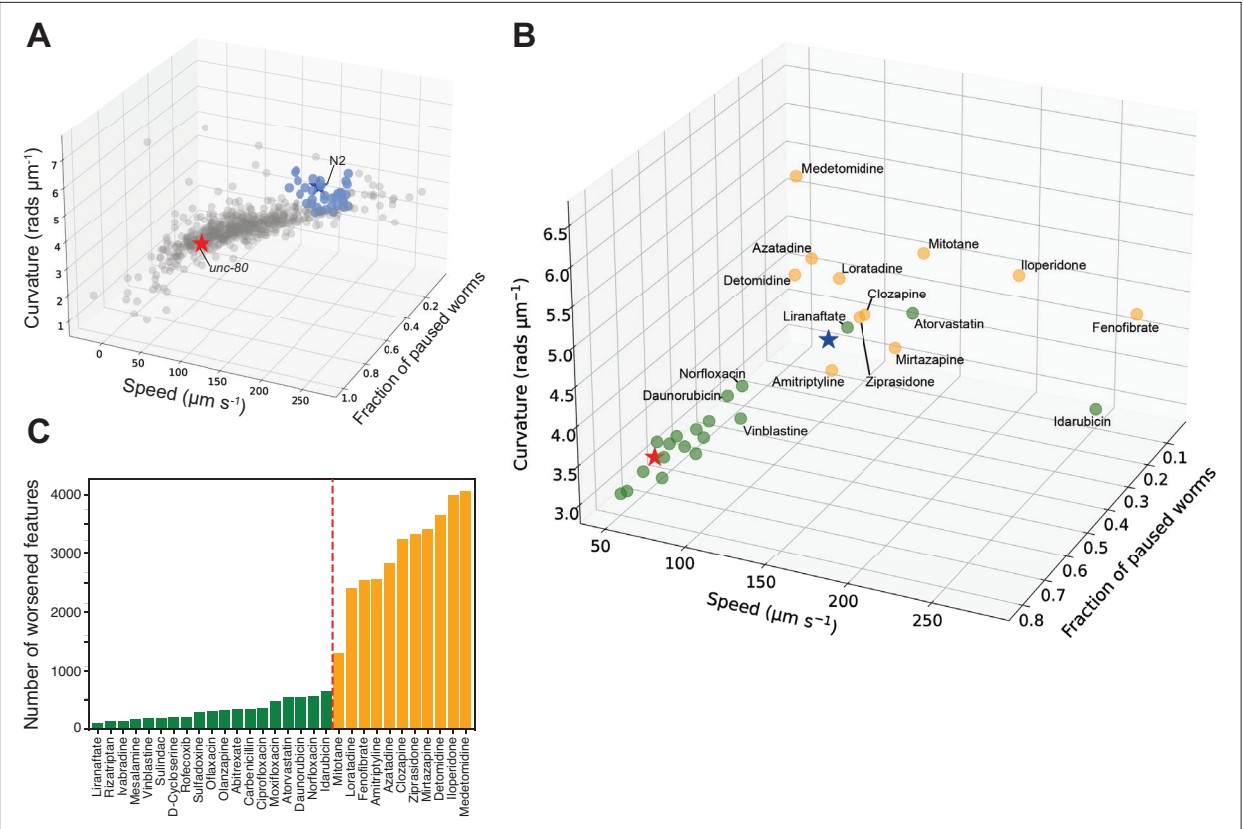

**Figure 5.** Drug repurposing screening. (**A**) Phenotypes of unc-80(syb1531) mutant (red star) and N2 (blue star) worms treated with 1% DMSO, and unc-80(syb1531) mutants treated with a library of 743 FDA-approved drugs at a concentration of 100 µM for 4 hr (circles). Each point represents an average of three well replicates, across three independent days of tracking (n=9 total). Blue points are the top 30 compounds that significantly improved all three of the core behavioural features, pushing the unc-80 mutant strain towards the control in phenospace. (**B**) Confirmation screen of the top 30 compounds identified in the initial library screen. Again unc-80(syb1531) and N2 DMSO treated controls are represented by red and blue stars, respectively, and each circular point represents unc-80(syb1531) treated with 100 µM compound for 4 hr. The 13 compounds coloured in the yellow lead to the worsening of >1000 behavioural features (see below). Liranaftate and atorvastatin both lead to a rescue of the core mutant phenotype with a low number of side effects. (**C**) Total number of behavioural 'side effects' following treatment of unc-80(syb1531) with the 30 compounds in the confirmation screen. Side effects are defined as features that are not significant between unc-80 mutants and wild-type N2 worms treated with 1% DMSO but where there is a significant difference between unc-80 mutants treated with a drug compared to N2. Red dashed line separates drug treatments that lead to a worsening of >1000 behavioural features that correspond to the points coloured in yellow in the 3D scatterplot.

The online version of this article includes the following figure supplement(s) for figure 5:

**Figure supplement 1.** Behavioural side effects across a reduced behavioural featureset.

We, therefore, defined a reduced set of core features to capture the unc-80 behavioural phenotype consisting of curvature, speed, and fraction of paused worms all during blue light stimulation (***Figure 4B–D***).

For the repurposing screen, we used a library of 743 FDA-approved drugs. We prepared tracking plates where each well contained a drug at a final concentration of 100 µM with 1% DMSO. Three adult worms were added to each well, incubated for 4 hr, and then tracked. The screen was repeated across three independent tracking days with three independent wells per day (nine total well replicates). Hits were defined as compounds that significantly improved all three of the core unc-80 features, meaning that treated worms were significantly different from unc-80 DMSO controls and that the direction of the effect was towards wild-type controls. These hit compounds shift unc-80 towards N2 in a three-dimensional phenotype space (***Figure 5A***). To test the utility of reducing the feature set in a screen compared to testing a larger feature set, we repeated this with larger feature sets. Testing larger feature sets reduced our power to detect differences. No hits were detected using a predefined set of 256 features (***Javer et al., 2018b***; ***Figure 4—figure supplement 1***).

We then performed a confirmation screen on 30 of the hit compounds from the initial screen with a larger number of replicates (24 wells per day over three independent days). 13 of the 30 hits were reproduced in the sense of showing significant differences from *unc-80* DMSO controls in the direction of wild-type, however, several showed a reduced effect size and were clustered around the *unc-80* DMSO controls (*Figure 5B*). Having a larger number of replicates, we also tested for 'side effects' in treated worms which we define as features where there was no significant difference between *unc-80* mutants and wild-type N2 animals but where there is a significant difference between *unc-80* mutants treated with a drug compared to wild-type N2 animals. About half of the confirmed hit compounds lead to more than 1000 side effects (*Figure 5C*) and are coloured yellow in the core *unc-80* phenotype space (*Figure 5B*). The same trend is observed when using 256 predefined features (*Javer et al., 2018b*; *Figure 5—figure supplement 1*).

Of the confirmed hit compounds with fewer than 1000 side effects, liranaftate and atorvastatin (Lipitor) lead to good rescue of the core *unc-80* behavioural phenotypes. Interestingly, both compounds act upon the mevalonate biosynthesis pathway. The main trunk of this pathway is conserved across species and converts acetyl-CoA to farnesyl diphosphate (*Rauthan and Pilon, 2011*). Atorvastatin is an inhibitor of HMG-CoA reductase that regulates lipid homeostasis (*Goldstein et al., 2006*) and catalyses the rate-limiting step of the mevalonate pathway. Liranaftate, an antifungal inhibitor of squalene epoxidase, shows the greatest behavioural improvement, with the fewest number of side effects. However, *C. elegans* lacks the branch of the mevalonate pathway responsible for converting squalene to cholesterol, instead relying on dietary cholesterol. In this case, the effect may be due to an off-target interaction on one of the other enzymes in the pathway.

Fenofibrate, a PPAR-alpha activator that increases lipolysis, is also a hit but has a large number of side effects. Given that it overshoots perfect rescue by shifting *unc-80* animals past wild-type controls in core phenospace, it could be that at a lower dose, it would provide good rescue with fewer side effects. Since UNC-80 is involved in localisation of the NACLN+ channel complex, we hypothesise that changes in membrane composition might be indirectly affecting the localisation or function of the channel complex.

## Discussion

We framed the problem of disease modelling as one of sampling and elucidating a model organism genotype-phenotype map that is connected to a human disease genotype-phenotype map through genetic conservation. However, both the genomic and phenomic spaces are large and high-dimensional, and so it was not clear that a systematic approach would identify useful phenologs for disease modelling. Using an initial set of genes that are diverse in function (but biased towards roles in neurons and muscles) and that lead to diverse symptoms when mutated in humans, we found that it is possible to systematically create and phenotype worm models of rare diseases using a uniform assay and protocol. Without adding additional perturbations or experimental conditions, we found that 23 out of the 25 disease models we made had detectable differences compared to the wild-type.

Of the disease models with detectable phenotypes, approximately half had strong phenotypes that would support high throughput phenotypic screens for candidate treatments. To test the feasibility of performing drug repurposing screens using the same uniform assay, we chose a worm model of UNC80 deficiency and, focussing on a reduced core phenotype, identified an initial list of 30 FDA-approved drugs that rescued the mutant phenotype. A confirmation screen with more replicates identified several compounds that still rescued the core disease model phenotype and did not have a large number of side effects. For UNC80 deficiency we chose a short treatment time of 4 hr to model treatment in patients where some developmental effects will have already occurred. Our approach is also compatible with longer treatments where animals at the first larval stage are added to tracking plates and allowed to develop in the presence of drugs. This might reveal other hit compounds and would be particularly interesting for mutants with developmental delays including some in this study (e.g. *cat-4*, *dys-1*, *Figure 1*, *gpb-2*, and *kcc-2*).

For this proof-of-principle study, we focused on the generation of large deletion alleles to help ensure the target gene is functionally disrupted. However, patient-specific missense mutations can also be introduced at conserved loci within genes and these mutations can lead to detectable behavioural phenotypes (see for example *Barlow et al., 2022*; *Wang et al., 2022*). There is also value in studying multiple alleles even for loss-of-function mutations. The large deletions we use here can

delete regulatory elements or non-coding RNAs that could lead to phenotypes not related to the modelled disease. Alternative methods of creating loss-of-function alleles such as introducing premature stop codons can also be done at the scale of dozens of strains (**Pu et al., 2023**). Furthermore, to study loss of function in essential genes, balanced heterozygous strains can be created where this makes sense for a particular disease model.

With a current discovery rate of >100 new genetic diseases each year, the number of diseases with little aetiological understanding is going to grow at a faster rate than biomedical and pharmaceutical research will find therapies (**Boycott et al., 2017**). Therefore, novel approaches that can systematically screen the effects of large numbers of genetic variants associated with different diseases are required. CRISPR genome editing in model organisms including *C. elegans* allows the creation of new disease models at the required rate and cost. We have shown that high-throughput worm tracking can identify screenable phenotypes and be used for repurposing screens. However, we do not yet know whether the phenotypes and repurposing hits will be useful for translation. It is possible that the phenotypes we detect are not direct results of key disease mechanisms but are instead secondary effects on connected systems. We expect that secondary effects are less likely to be driven by conserved mechanisms but this can only be assessed on a case-by-base basis. Similarly, the 'side effects' we detect in worms may not be predictive in humans. Still, we think quantifying side effects will be useful in some cases. For example, if a disease model is hyperactive and gross motility were the only measure used in a screen, compounds that are broadly toxic would be detected as hits. This kind of broad toxicity would result in a large number of side effects and could be avoided. Ultimately, the translatability of hits in invertebrate screens can only be determined in clinical trials, and choosing hits from phenotypic screens to pursue clinically will require judgement that takes into account the full picture available from multiple data sources including the strength of a rescue, the putative drug mechanism of action, knowledge of disease biology, and an assessment of the risk of testing a given drug in patients. We are optimistic that we will see more human trials based on screens in diverse small animal models.

## Materials and methods

### Mutant generation

CRISPR guide RNAs were designed to target large deletions (>1000 bp) that start close to the start codon and excise several exons from the gene in order to give high confidence of loss of function. For exceptionally large genes (e.g. *dys-1* or *mpz-1*) the entire protein-coding region was excised. Mutants were designed and made by SunyBiotech in an N2 background.

### Worm preparation

All strains were cultured on Nematode Growth Medium at 20 °C and fed with *E. coli* (OP50) following standard procedure (**Stiernagle, 2006**). Synchronised populations of young adult worms for imaging were cultured by bleaching unsynchronised gravid adults, and allowing L1 diapause progeny to develop for 2.5 d at 20 °C (detailed protocol: https://dx.doi.org/10.17504/protocols.io.2bzgap6). Several strains were developmentally delayed and were allowed to grow for longer before imaging. *cat-4*(*syb1591*), *gpb-2*(*syb1577*), *kcc-2*(*syb2673*), and *unc-25*(*syb1651*) were allowed to develop for 3.5 and *dys-1*(*syb1688*), **Figure 1** (*syb1562*), and *pink-1*(*syb1546*) were allowed to develop for 5.5 d prior to imaging. On the day of imaging, young adults were washed in M9 (detailed protocol: https://dx.doi.org/10.17504/protocols.io.bfqbjmsn), transferred to the imaging plates (three worms per well) using a COPAS 500 Flow Pilot (detailed protocol: https://dx.doi.org/10.17504/protocols.io.bfc9jiz6), and returned to a 20 °C incubator for 3.5 hr. Plates were then transferred onto the multi-camera tracker for another 30 min to habituate prior to imaging (detailed protocol: https://dx.doi.org/10.17504/protocols.io.bsicncaw).

For drug repurposing experiments, the MRCT FDA-approved compound library (Catalog No.L1300) was supplied pre-dissolved in DMSO by LifeArc (Stevenage, UK). The day prior to tracking, imaging plates were dosed with the compound library to achieve a final well concentration of 100 μM prior to seeding with bacteria (see below for details). Plates were left to dry (~30 min), before being stored in the dark at room temperature overnight. Following the methods described above, age-synchronised young adult *unc-80*(*syb1531*) worms were dispensed into the imaging plate wells and incubated at 20 °C for 4 hr before tracking. The behaviour of mutant worms dosed with the drugs

was then compared to wild-type N2 and *unc-80*(*syb1531*) worms (also age-synchronised young adults) dispensed into the wells of the same tracking plates dosed with an identical volume (1% w/v) of DMSO only (detailed protocol: https://dx.doi.org/10.17504/protocols.io.5jyl8p5yrg2w/v1).

### Plate preparation
Low peptone (0.013%) nematode growth medium (detailed protocol: https://dx.doi.org/10.17504/protocols.io.2rcgd2w) was prepared as follows: 20 g agar (Difco), 0.13 g Bactopeptone, and 3 g NaCl were dissolved in 975 mL of milliQ water. After autoclaving, 1 mL of 10 mg/mL cholesterol was added along with 1 mL CaCl2 (1 M), 1 mL MgSO4 (1 M), and 25 mL KPO4 buffer (1 M, pH 6.0). Molten agar was cooled to 50–60°C and 200 μL was dispensed into each well of 96-square well plates (Whatman UNIPLATE: WHAT-77011651) using an Integra VIAFILL (detailed protocol: https://dx.doi.org/10.17504/protocols.io.bmxbk7in). Poured plates were stored agar-side up at 4 °C until required.

One day prior to imaging, plates were placed without lids in a drying cabinet to lose 3–5% of weight by volume. Wells were then seeded with 5 μL OP50 (OD600 1.0) using an Integra VIAFILL dispenser, and plates were stored with lids (WHAT-77041001) on at room temperature overnight.

### Image acquisition
All videos were acquired and processed following methods previously described (*Barlow et al., 2022*). In brief, videos were acquired at 25 frames per second in a room with a nominal temperature of 20 °C using a shutter time of 25 ms and a resolution of 12.4 μm/px. Three videos were taken sequentially: a 5 min pre-stimulus video, a 6 min blue light recording with three 10 s blue light pulses starting at 60, 160, and 260 s, and a 5 min post-stimulus recording. The script for controlling recording timings and photostimulation was made using LoopBio's API for their Motif software(https://github.com/loopbio/python-motifapi; *nzjrs, 2024*).

### Image processing and feature extraction
Videos were segmented and tracked using Tierpsy Tracker (*Javer et al., 2018a*). After the segmentation of worm skeletons, our previously described convolutional neural network classifier was used to exclude non-worm objects from being classified during feature extraction (*Barlow et al., 2022*). Skeletons that did not meet the following criteria were removed from the analysis: 700–1300 μM length, 20–200 μM width. As an additional quality control measure, we used Tierpsy Tracker's viewer to mark wells with visible contamination, agar damage, compound precipitation, or excess liquid as 'bad,' and exclude these wells from downstream analysis.

Following tracking, we extracted a previously-defined set of 3076 behavioural features for each well in each of the three videos (pre-stimulus, blue light, and post-stimulus) (*Javer et al., 2018b*). Feature values are averaged over tracks to produce a single feature vector for each well.

### Statistical analysis
Statistically significant differences in the pre-stimulus, post-stimulus and blue-light behavioural feature sets extracted from each disease model strain compared to our N2 reference were calculated using block permutation t-tests (https://github.com/Tierpsy/tierpsy-tools-python/blob/master/tierpsytools/analysis/statistical_tests.py; *Tierpsy, 2021*). Python (version 3.8.5) was used to perform the analysis, using n=10,000 permutations that were randomly shuffled within, but not between, the independent days of image acquisition for each strain to control for day-to-day variation in the experiments. The *p*-values were then corrected for multiple comparisons using the Benjamini-Yekutieli procedure to control the false discovery rate at 5% (*Benjamini and Yekutieli, 2001*).

Heatmaps, cluster maps, and principal component analysis of the extracted feature sets for each disease model strain compared to our N2 reference were calculated using in-built Seaborn (version 0.11.2) packages (*Waskom, 2021*). All scripts used for statistical analysis and the generation of figures are available at: https://github.com/Tom-OBrien/Systematic-creation-and-phenotyping-of-Mendelian-disease-models-in-*C.elegans* (copy archived at *O'Brien, 2024*).

### Materials availability
Worm strains will be made available through the *C. elegans* Genetics Centre and are available upon request from the corresponding author.

## Acknowledgements

This project has received funding from the European Research Council (ERC) under the European Union's Horizon 2020 research and innovation programme (Grant agreement No. 714853) and was supported by the Medical Research Council through grant MC-A658-5TY30.

## Additional information

### Funding

| Funder | Grant reference number | Author |
|---|---|---|
| European Research Council | 714853 | Thomas J O'Brien<br>Ida L Barlow<br>Luigi Feriani<br>André EX Brown |
| Medical Research Council | MC-A658-5TY30 | Thomas J O'Brien<br>Ida L Barlow<br>Luigi Feriani<br>André EX Brown |

The funders had no role in study design, data collection and interpretation, or the decision to submit the work for publication.

### Author contributions

Thomas J O'Brien, Data curation, Software, Formal analysis, Investigation, Visualization, Methodology, Writing – original draft; Ida L Barlow, Data curation, Software, Formal analysis, Validation, Investigation, Visualization, Methodology; Luigi Feriani, Data curation, Software, Formal analysis, Visualization; André EX Brown, Conceptualization, Supervision, Funding acquisition, Visualization, Writing – original draft, Project administration, Writing – review and editing

### Author ORCIDs

André EX Brown (ID) https://orcid.org/0000-0002-1324-8764

Reviewer #3 (Public review): https://doi.org/10.7554/eLife.92491.4.sa1
Author response https://doi.org/10.7554/eLife.92491.4.sa2

## Additional files

### Supplementary files

Supplementary file 1. Table indicating the unique Wormbase and Ensembl database accession numbers for every *C. elegans* gene in our panel of disease model mutants. Alongside denoting which (and how many) gene orthology programs predict that the worm gene is an orthologous to a human gene, and the genetic similarity and Blast-E score for each *C. elegans* gene.

Supplementary file 2. Document with a 1-page summary of each disease model created including basic genetic information, human and worm gene names, phenotypic fingerprint, and box plots for selected features comparing the disease model to wild-type worms.

Supplementary file 3. Summary of the associated human ortholog(s), predicted functional class, and key associated human disease phenotypes (Human Phenotype Ontology database) for each *C. elegans* disease model mutant.

MDAR checklist

### Data availability

Data and code to generate plots in the figures is available at https://doi.org/10.5281/zenodo.12684118.

The following dataset was generated:

| Author(s) | Year | Dataset title | Dataset URL | Database and Identifier |
|---|---|---|---|---|
| O'Brien T, Barlow I, Feriani L, Brown A | 2024 | Systematic creation and phenotyping of Mendelian disease models in *C. elegans*: towards large-scale drug repurposing | https://doi.org/10.5281/zenodo.12684118 | Zenodo, 10.5281/zenodo.12684118 |

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
