## [Editor Report · eLife Assessment]

This **important** study provides proof of principle that *C. elegans* models can be used to accelerate the discovery of candidate treatments for human Mendelian diseases by detailed high-throughput phenotyping of strains harboring mutations in orthologs of human disease genes. The data are **compelling** and support an approach that enables the potential rapid repurposing of FDA-approved drugs to treat rare diseases for which there are currently no effective treatments. The work will be of interest to all geneticists.

---

## [Referee Report · Reviewer #3 (Public review)]

In this study, O'Brien et al. address the need for scalable and cost-effective approaches to finding lead compounds for the treatment of the growing number of Mendelian diseases. They used state-of-the-art phenotypic screening based on an established high-dimensional phenotypic analysis pipeline in the nematode *C. elegans*.

First, a panel of 25 *C. elegans* models was created by generating CRISPR/Cas9 knock-out lines for conserved human disease genes. These mutant strains underwent behavioral analysis using the group's published methodology. Clustering analysis revealed common features for genes likely operating in similar genetic pathways or biological functions. The study also presents results from a more focused examination of ciliopathy disease models.

Subsequently, the study focuses on the NALCN channel gene family, comparing the phenotypes of mutants of nca-1, unc-77, and unc-80. This initial characterization identifies three behavioral parameters that exhibit significant differences from the wild type and could serve as indicators for pharmacological modulation.

As a proof-of-concept, O'Brien et al. present a drug repurposing screen using an FDA-approved compound library, identifying two compounds capable of rescuing the behavioral phenotype in a model with UNC80 deficiency. The relatively short time and low cost associated with creating and phenotyping these strains suggest that high-throughput worm tracking could serve as a scalable approach for drug repurposing, addressing the multitude of Mendelian diseases. Interestingly, by measuring a wide range of behavioural parameters, this strategy also simultaneously reveals deleterious side effects of tested drugs that may confound the analysis.

Considering the wealth of data generated in this study regarding important human disease genes, it is regrettable that the data is not made accessible to researchers less versed in data analysis methods. This diminishes the study's utility. It would have a far greater impact if an accessible and user-friendly online interface were established to facilitate data querying and feature extraction for specific mutants. This would empower researchers to compare their findings with the extensive dataset created here.

Another technical limitation of the study is the use of single alleles. Large deletion alleles were generated by CRISPR/Cas9 gene editing. At first glance, this seems like a good idea because it limits the risk that background mutations, present in chemically-generated alleles, will affect behavioral parameters. However, these large deletions can also remove non-coding RNAs or other regulatory genetic elements, as found, for example, in introns. Therefore, it would be prudent to validate the behavioral effects by testing additional loss-of-function alleles produced through early stop codons or targeted deletion of key functional domains.

Comments on revisions:

In this final round of revisions, the authors have improved their manuscript and provide useful information about analysis procedures and code and updated figures.

---

## [Author Response]

The following is the authors’ response to the previous reviews.

This important study provides proof of principle that *C. elegans* models can be used to accelerate the discovery of candidate treatments for human Mendelian diseases by detailed high-throughput phenotyping of strains harboring mutations in orthologs of human disease genes. The data are compelling and support an approach that enables the potential rapid repurposing of FDA-approved drugs to treat rare diseases for which there are currently no effective treatments. The authors should provide a clearer explanation of how the statistical analyses were performed, as well as a link to a GitHub repository to clarify how figures and tables in the manuscript were generated from the phenotypic data.

We have amended our description of the statistical analysis in the materials and methods section of the manuscript. We have also updated the GitHub repository link to a dedicated repository for this study, this contains all of the code needed to generated all the figures made from the phenotypic data provided. Additionally, we have updated the Zenodo repository to contain both the code and datasets within the same file.

We have also updated the GitHub repository link to a dedicated repository for this manuscript, that contains all of the code needed to generate all figures from the phenotypic data provided. Additionally, we have updated the Zenodo repository link to contain both the code and datasets within the same folder structure.

**Recommendations for the authors:**

**Reviewer #1 (Recommendations for the authors):**
The authors have responded to previous review to improve the presentation of the work. The paper more than meets publication standards.

No response required.

**Reviewer #2 (Recommendations for the authors):**
The authors have addressed all of my questions and concerns. I'm happy to see this updated paper of record.

No response required.

**Reviewer #3 (Recommendations for the authors):**
Regarding the interactive heatmapThe html version and the panel in Figure 2C appear not to coincide visually. Maybe the features are ordered in a different way?

The html version of Figure 2C is for the entire feature set extract per strain and not the condensed Tierpsy256 set shown in the panel figure. We have now remade this figure to show this reduced feature set (aligning with what is shown in Figure 2C) and included both versions of the interactive heatmaps as static html files within the same repository.

Regarding data accessibility overallMore generally, the html file does not address my initial concern about the accessibility of the data to non-experts. Making the full dataset available was a necessary first step, but the hermetic nature of its format and the lack of a simple way to query the data remains an issue for me that limits the usefulness of this data to the broadest audience.

We agree, but unfortunately do not currently have the resources to build a public-facing database to facilitate this.